# Renewable Energy Potential: Second-Generation Biomass as Feedstock for Bioethanol Production

**DOI:** 10.3390/molecules29071619

**Published:** 2024-04-04

**Authors:** Chidiebere Millicent Igwebuike, Sary Awad, Yves Andrès

**Affiliations:** IMT Atlantique, GEPEA, UMR CNRS 6144, 4 Rue Alfred Kastler, F-44000 Nantes, France; chidieberem.igwebuike@gmail.com (C.M.I.); yves.andres@imt-atlantique.fr (Y.A.)

**Keywords:** feedstock, biomass conversion, fermentation, biofuels, bioethanol, projects

## Abstract

Biofuels are clean and renewable energy resources gaining increased attention as a potential replacement for non-renewable petroleum-based fuels. They are derived from biomass that could either be animal-based or belong to any of the three generations of plant biomass (agricultural crops, lignocellulosic materials, or algae). Over 130 studies including experimental research, case studies, literature reviews, and website publications related to bioethanol production were evaluated; different methods and techniques have been tested by scientists and researchers in this field, and the most optimal conditions have been adopted for the generation of biofuels from biomass. This has ultimately led to a subsequent scale-up of procedures and the establishment of pilot, demo, and large-scale plants/biorefineries in some regions of the world. Nevertheless, there are still challenges associated with the production of bioethanol from lignocellulosic biomass, such as recalcitrance of the cell wall, multiple pretreatment steps, prolonged hydrolysis time, degradation product formation, cost, etc., which have impeded the implementation of its large-scale production, which needs to be addressed. This review gives an overview of biomass and bioenergy, the structure and composition of lignocellulosic biomass, biofuel classification, bioethanol as an energy source, bioethanol production processes, different pretreatment and hydrolysis techniques, inhibitory product formation, fermentation strategies/process, the microorganisms used for fermentation, distillation, legislation in support of advanced biofuel, and industrial projects on advanced bioethanol. The ultimate objective is still to find the best conditions and technology possible to sustainably and inexpensively produce a high bioethanol yield.

## 1. Introduction

Currently, emphasis has shifted to the use of renewable sources of energy both in the developed and developing countries of the world. This is due to the steady growth in population and industrialization [1], the decline in known reserves, the rising uncertainty of petroleum supplies as a result of increasing demand, and concerns over climate change (greenhouse gas emissions and global warming), which are linked to the use of fossil fuels [2]. In a bid to satisfy the world demand for energy, lower fossil fuel consumption, reduce the emissions of CO_2_, and maintain or develop agricultural activities (by utilizing bio-resources for energy, food, and material), governments across the world are encouraging the exploitation of renewable resources and energies such as biomass, wind, solar and hydroelectricity [3]. Technological advancement in biofuel production supports greener modes of transportation, contributing to a sustainable energy future. Among the available alternative energy sources, biofuels stand out in their general compatibility with existing liquid transport fuels. They are considered a viable alternative to fossil-based fuels. Bioethanol, a type of liquid biofuel, plays a vital role in the global energy transition by reducing greenhouse gas emissions and enhancing energy security. Its use can promote rural development, drive economic growth, and satisfy international targets for renewable energy. To produce bioethanol, lignocellulosic waste plant biomass could be exploited. It is a second-generation feedstock and it is regarded as the most abundantly available, renewable, and inexpensive energy source for the production of bioethanol [4], owing to its high holocellulose composition [5]. This biomass stands out because it does not serve as food for human consumption. Hence, there is no competition in the food market compared to the case of first-generation biomass (e.g., sugarcane and maize). Lignocellulose accounts for more than 90% of worldwide plant biomass production. It amounts to approximately 200 billion tons per year [2].

Lack of knowledge about the importance of agricultural wastes, inadequate finance for the purchase of needed facilities/equipment, and lack of technical know-how or skills for biofuel production are the major causes of incessant disposal of agricultural waste (e.g., cassava peels, sugar beet pulp, *Ulva lactuca*, sugarcane bagasse, corn straw, etc.) in the environment, especially in the rural areas of developing countries where agricultural activities are prevalent, and these wastes are usually rich in cellulose and hemicellulose components and can be broken down into simpler components for the production of useful fuels. The various activities practiced by humans in the disposal of these wastes have caused the deterioration of the environment and the loss of its aesthetic value. They have also caused several health challenges. The conversion of these wastes to useful fuels like bioethanol can help to reduce the occurrence of these environmental pollutants in our environment and also meet the energy demand of the populace. Notwithstanding the advantages inherent in the utilization of lignocellulosic biomass for bioethanol production, there exists a drawback due to the presence of lignin in the cell wall, which confers upon the material its recalcitrance nature and renders it not easily degradable. As a result of this, such material needs to be subjected to different treatment processes to release fermentable sugars. This review, therefore, aims to provide insights and, at the same time, explore the current state of the research on bioethanol production and identify the key challenges and opportunities for advancing the technology.

The study strategy involved conducting a comprehensive literature search using the Google Search Engine and relevant academic databases, such as ScienceDirect, Google Scholar, Researchgate, and PubMed. Keywords and phrases related to second-generation biomass, bioethanol production, and relevant technologies were used to search for potential articles, and the search results were screened based on titles and abstracts to identify relevant articles. Full-text articles were then reviewed to determine their eligibility for inclusion in this review. The quality of the included articles was evaluated based on the credibility of the reports/findings, and key insights were identified and analyzed.

## 2. Biomass and Bioenergy

For ages, humans have relied on the use of traditional bioenergy. Over 85% of biomass energy is being utilized as solid fuels for heating, cooking, and lighting at present, but these methods have low efficiency. Fuelwood and charcoal are categorized as traditional bioenergy, which only provides heat, and are said to dominate the consumption of bioenergy, especially in the developing world where about 95% of national energy consumption depends on biomass. This biomass will keep on being an essential source of bioenergy in many parts of the world. Up to now, wood fuels stand as the main source of bioenergy across different regions, as they offer energy security services for large divisions of society. Modern bioenergy depends on efficient conversion technologies for domestic use and utilization at both industrial scales and small businesses. The significance of bioenergy is demonstrated worldwide. In North America, the use of ethanol derived from corn reduces reliance on fossil fuels [6]; in Scandinavia, district heating using biomass reduces carbon emissions [7]; in Africa, the use of biofuels for off-grid power improves access to energy [8]; and in Southeast Asia, palm oil biodiesel industry tackles climate change while driving economic growth [9]. Biomass could be processed into more convenient energy carriers, such as liquid fuels (biodiesel, bioethanol, bio-oil), solid fuels (wood chips, firewood, charcoal, briquettes, pellets), gaseous fuels (hydrogen, synthesis gas, biogas), or heat directly from the production process [10]. The combustion of biomass is regarded as a carbon-neutral process since the carbon dioxide emitted has been absorbed by the plants from the atmosphere beforehand. So far, organic wastes and residues are the major biomass sources; nevertheless, energy crops such as poplar, willow, and eucalyptus are gaining significance and market share. Biomass resources comprise wood wastes from industry and forestry, agricultural residues, residues from paper and food industries, animal manure, dedicated energy crops, sewage sludge, municipal green wastes, starch crops (wheat, corn), sugar crops (beet, sugarcane, sorghum), oil crops (oilseed rape, soy, sunflower, palm oil, jatropha), and grasses (Miscanthus) [11].

At present, biomass constitutes only a small fraction of the total carbon use despite often being applied as raw material and a source of energy. Its use is limited to the large-scale production of bioethanol and low-volume products. In the coming years, there is expected to be a continual shift from the present fossil-based to a future bio-based carbon economy, thus causing a gradual effect on all process industries. There will be a constant transition to more complex bio-renewable feedstock, such as algae, agricultural residues, green plants, industrial wastes, or wood, and eventually, bio-based products will replace the petrochemical product tree. This transition is not to be regarded as a threat but a chance to redesign the industrial value chain from renewable material sources to new products and for this to be achieved, the rich molecular structure of renewable biomaterials is to be greatly exploited [12].

## 3. Structure and Composition of Lignocellulosic Biomass

Lignocellulosic biomass is majorly composed of three polymers: cellulose, hemicellulose, and lignin, including a few amounts of protein, pectin, ash, and extractives. The proportion of these components can differ from one biomass species to another. For instance, hardwoods have more amounts of cellulose, while leaves and wheat straws have more hemicellulose. The proportion of these components is also not often the same within a single plant species due to the stage of growth, age, and certain other conditions. These polymers are linked with each other in a hetero-matrix to different degrees and varying relative compositions depending on the type, species, and source of the biomass material [13]. The relative amounts of cellulose, hemicellulose, and lignin are, among others, important determining factors in identifying the suitability of plant species for use as energy crops [14]. Figure 1 gives the structural presentation of lignocellulosic biomass.

Cellulose is the most abundant fraction in lignocellulosic materials and represents about 40–50% of the biomass by weight. It is a polymer of glucose, comprising linear chains of (1,4)-D-glucopyranose units, whereby the units are linked C_1_–C_4_ oxygen bridges with the removal of water in the β-configuration (β-glycosidic bonds) with an average molecular weight of around 100,000 [14]. Cellulose is a white solid material that may exist either in crystalline or amorphous states. The crystalline state of cellulose confers its ability to be resistant to chemical attack and degradation. The high strength of cellulose fibers is a consequence of the hydrogen bonding that exists between cellulose molecules [15].

After cellulose, hemicellulose ranks as the second largest carbohydrate in the world. Daily production per person was estimated to be about 20 kg, and about 45,000 million tons are produced on a yearly basis [16]. Hemicellulose can be categorized into four general classes of structurally different cell wall polysaccharide types: xylans, mannans, β-glucans with mixed linkages, and xyloglucans. They exist in structural variations differing in side-chain types, distribution, localization, and/or types and distribution of glycoside linkages in the macro-molecular backbone [17]. The hemicellulose fraction represents 20–40% of the biomass by weight. It is a mixture of polysaccharides that comprises mostly sugars, like glucose, xylose, arabinose, mannose, methylglucuronic, and galacturonic acids. Hemicellulose has an average molecular weight of <30,000 [14]. In appearance, hemicelluloses are also white solid materials and are amorphous in nature [15].

Lignin refers to a group of amorphous that have high molecular weight (over 10,000), are cross-linked, and are chemically related complex polymer compounds that form an essential part of the secondary plant cell wall. It is rich in aromatic subunits and relatively hydrophobic. It hinders the free access of cellulolytic enzymes due to its cross-linkages with other components of the cell wall. It has a very slow rate of decomposition but contributes a significant aspect to the materials that form humus [18]. Lignin comprises three basic monomers: *p*-coumaryl alcohol, coniferyl alcohol, and sinapyl alcohol. All three lignin monomers are found in straws and grasses. Coniferyl alcohol is found in softwoods (gymnosperms e.g., cycads and conifers), while both coniferyl alcohol (50–75%) and sinapyl alcohol (25–50%) are present in hardwoods (dicotyledonous angiosperms) [15].

Lignocellulosic materials have great potential to generate second-generation biofuels, including bio-sourced materials and chemicals, without negatively impacting the world’s food security. However, the main drawback of these materials’ valorization is their recalcitrance to enzymatic hydrolysis due to the heterogeneous multi-scale structure of plant cell walls. The factors affecting the recalcitrance of these materials are strongly interconnected and not easily dissociated. These factors can be classified into chemical factors (composition and content in lignin, hemicelluloses, acetyl groups) and structural factors (cellulose crystallinity, cellulose specific surface area, degree of polymerization, pore size, and volume) [19]. The structural composition of different lignocellulosic biomass is given in Table 1.

**Figure 1 molecules-29-01619-f001:**
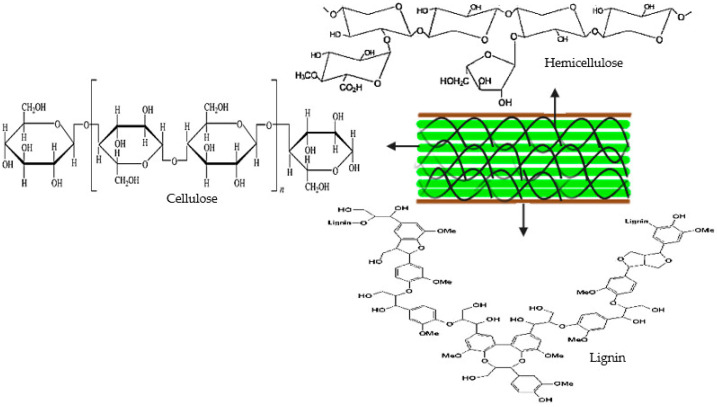
Structure and composition of lignocellulosic biomass [20,21].

## 4. Groups of Biofuels

Depending on the processing before utilization, biofuels can be classified into primary and secondary biofuels (Figure 2).

### 4.1. Primary Biofuels

Primary biofuels are used in their unprocessed form, i.e., the organic material is used in its natural form, as harvested. Examples of such primary biofuels include forest residues, pellets, wood chips, firewood, agricultural crops, and animal fats. They are primarily used for cooking, heating, agricultural needs, and the production of electricity in small- and large-scale industrial applications. They are common in developing countries. Primary biofuels are also referred to as traditional biomass. The field of application of this biofuel is small, and it does not require processing resource expenses. Energy derived from traditional biomass accounted for about 9% of all energy consumed globally for the year 2013 [37,38,39,40,41].

### 4.2. Secondary Biofuels

Secondary biofuels are processed materials. They are produced by processing biomass (the primary biofuel). Secondary biofuels can be in the form of solids (charcoal), liquids (bio-oil, ethanol, biodiesel), or gases (hydrogen, biogas, synthesis gas), which can be used in a wider range of applications, such as high-temperature industrial processes and transport, to substitute fossil fuel. Based on the type of raw materials, the historical sequence of the fuel’s appearance on the world energy market and the processing technology employed in production, secondary biofuels can further be divided into three generations: first-generation biofuels, second-generation biofuels, and third-generation biofuels [37,38,39,40,42]. Table 2 gives some benefits and issues associated with the distinct biofuel generations.

#### 4.2.1. First-Generation Biofuels

First-generation biofuels are fuels that have been produced from biomass that are generally edible, for example, corn and sugarcane. Sugar beets, barley, potato wastes, and whey are also some of the marginal feedstocks that are used or considered to produce first-generation bioethanol [43]. From the environmental and economic outlook, sugarcane is an ideal feedstock for the production of ethanol, but it is limited to certain regions due to soil and weather conditions [44]. First-generation biofuels can help to enhance domestic energy security and can offer some CO_2_ benefits [45]. However, the major concern about these biofuels is their inefficiency and sustainability since the viability of the production of such biofuel is questionable as a result of the conflict with food supply [46], including sourcing of feedstock, land use, deforestation, and the impact it may have on biodiversity. Nowadays, first-generation biofuel production is commercial, with an annual production of about 50 billion liters. Biofuels (first generation) like bioethanol, biodiesel, and biogas are categorized by their ability to be blended with petroleum-based fuels, combusted in existing internal combustion engines, and distributed through existing infrastructure, or by their use in existing alternative vehicle technology, such as natural gas vehicles or flexible-fuel vehicles (FFVs) [45]. First-generation biofuels compete with food and accrue high costs of production. Certain crops and foodstuffs have become expensive due to the fast expansion of global biofuel production from sugar, grain, and oilseed crops; hence, with these drawbacks, there is, therefore, a need to search for non-edible biomass for biofuel production [39].

#### 4.2.2. Second-Generation Biofuels

Second-generation biofuels are produced from lignocellulosic materials, such as corn straw, wheat straw, sugarcane bagasse, sugar beet pulp, cassava peels, and switchgrass, including softwood and hardwood. They rely on the use of biomass that is not suitable for being used as food (non-edible biomass). Second-generation biofuels comprise either plants that are mainly grown for energy production, i.e., bioenergy crops on marginal lands (lands that are unsuitable for food production) or non-edible parts of crops and forest trees, which should be processed efficiently for bioenergy by improving existing technology [47]. Lignocellulose, which makes up the cell walls of plant biomass, is divided into three main components, which include cellulose (30–50%), hemicellulose (10–40%), and lignin (5–20%) [14]. Different authors have certain values for all of these components; however, the extraction process of a particular component, particularly cellulose, is somewhat difficult [40]. The development of second-generation biofuels is generally seen as a sustainable response to the rising controversy surrounding first-generation biofuels [48]. The potential for bioethanol production will be influenced by the chemical composition of the organic compounds involved [37]. Second-generation liquid biofuels are commonly produced by two different methods, which include biological or thermochemical processing from lignocellulosic agricultural biomass. The main benefit of producing second-generation biofuels from inedible feedstock is that it curtails the direct food versus fuel competition connected with first-generation biofuels. Furthermore, in comparison with first-generation biofuels, second-generation biofuels are associated with an increase in land use efficiency and reduced pressure on biodiversity [39]. According to Markus et al. [49], the cell walls of plant biomass represent one of the most abundant renewable resources on earth. At present, only 2% of this biomass is utilized by man in spite of its abundance. This calls for the need to research the feasibility of utilizing plant cell walls in the production of inexpensive biofuels. The major drawback in the use of lignocellulosic materials is the recalcitrance of cell walls to degrade efficiently into simple fermentable sugars. The addition of wall structure-altering agents or the manipulation of the wall polysaccharide biosynthetic machinery should make the tailoring of wall composition and architecture possible in order to improve sugar yields for biofuel production. However, the main challenge is that the study of biosynthetic machinery and its regulation is still in its early stages.

#### 4.2.3. Third-Generation Biofuels

Third-generation biofuels commonly refer to biofuels produced from algal biomass [50]. Microalgae biomass as a candidate for biofuel production is becoming popular owing to its rapid growth rate, high lipid and starch content, ease of cultivation [51], low land usage, and high carbon dioxide absorption [50]. It offers a potential solution to one of the pressing issues faced by modern societies today (the development of renewable energy for transportation) owing to its high surface biomass productivity, ability to grow on marginal lands, and efficient conversion of solar energy to chemical energy [52]. Algae has the ability to produce higher yields with lesser input resources than other biomass; hence, it is separately classified from second-generation biofuels. In terms of the potential of fuel production, with regard to quantity or diversity, no feedstock can compete with algae. The two attributes of algae with respect to the diversity of fuel it can produce include the following; (i) it produces oil that can be easily refined into diesel or certain gasoline components and (ii) it can be genetically manipulated to produce fuels like ethanol, butanol, diesel, and gasoline. Algae is able to produce outstanding yields and has produced about 9000 gallons of biofuel per acre, and this is about 10-fold more than what the best traditional feedstock has been known to produce. There has been a suggestion by those who work closely with algae that yields as high as 20,000 gallons per acre are achievable. Notwithstanding, algae biomass has some drawbacks, and one of these drawbacks is that it requires large volumes of water, phosphorus, and nitrogen to grow, even when grown in wastewater. Also, biofuels produced from algae tend to be less stable than those produced from other sources. The reason for this is that the oil present in algae tends to be highly unsaturated, especially at high temperatures, and, hence, more liable to degradation [50]. Furthermore, microalgae biofuels are not yet commercially sustainable. There are still challenges with regard to the improvement of microalgae strains and cultivation technologies [52].

## 5. Bioethanol as an Energy Source

In recent years, there has been an increase in the global production and use of biofuels, for example, from 18.2 billion liters to 60.6 billion liters to 162 billion liters in 2000, 2007, and 2019, respectively, with about 85% of this being bioethanol [2,53]. Bioethanol can be produced from a variety of cheap substrates, and it is reported to be one of the important and most widely used liquid biofuels worldwide, especially in transportation [8]. On average, ethanol has about 33% less energy content than gasoline [54]. Depending on the feedstock used, it has been estimated that bioethanol is able to lower the emission of greenhouse gas by approximately 30–85% compared to gasoline [2,55]. An 85% ethanol–15% gasoline blend can cut greenhouse gas emissions by 60–80%. A mixture of 10% ethanol and 90% gasoline can cut emissions by up to 8% [56]. Bioethanol and biodiesel are the two most common biofuels. While bioethanol is produced by the fermentation of biomass rich in carbohydrates, biodiesel can be produced from animal fats, vegetable oils, algae, or recycled cooking greases. Ethanol can be used as a fuel additive and biodiesel can be used either in its pure form or as a diesel additive to fuel a vehicle [57].

Ethanol has been described as being perhaps the oldest product obtained through traditional biotechnology [58]. The use of ethanol as a fuel for motors can be traced back to the days of the Model T. The first set of people to identify that the abundant sugars and starches present in plant biomass could be cheaply and easily converted to renewable biofuel were Henry Ford and Alexander Graham Bell. In the year 2016, the United States was the highest producer of ethanol worldwide, producing almost 60% of the global production [59], and this position is maintained to date.

Bioethanol finds application in the transportation sector, beverage and pharmaceutical industries, and electricity generation. Residues from bioethanol production could be used to produce thermal energy, valuable chemicals, and fertilizers. Bioethanol is a possible alternative to fossil-based transportation fuels because it has broader flammability limits, higher octane number, higher heat of vaporization, and higher flame speeds, and these characteristics allow for a shorter burn time, higher compression ratio, and leaner burn engine, which ultimately results in an advantage in theoretical efficiency over gasoline in an internal combustion engine [60,61]. Mixing ethanol with petrol for transportation boosts the performance of the latter. It also enhances fuel combustion in vehicles, lowering the release of unburned hydrocarbons, carbon monoxide, and carcinogens. Nevertheless, the combustion of ethanol also leads to a heightened reaction with nitrogen in the atmosphere, which can cause a marginal increase in nitrogen oxide gases. Compared to petrol, ethanol contains only trace amounts of sulfur. So, when ethanol is mixed with petrol, it will help lower the fuel’s sulfur content and, hence, reduce the emissions of sulfur oxide, which is a major component of acid rain and a carcinogen [38].

A combination of sugar beets and wheat is generally used in the production of EU bioethanol. It has been projected that bioethanol production in the EU probably has a greater potential than biodiesel; this is coming from the estimated abundant supplies and production potential for sugar beets and cereals, but the cost of production of EU biofuels is a consequence of high-priced internal feedstock compared to fossil fuels and remains a major barrier to the market-based expansion of EU biofuel production, especially for bioethanol [62]. At present, France is a front-runner in the EU’s attempt to enhance the use of ethanol, accounting for 2% of global production, primarily from wheat and sugar beet [61,63], making France the top producer of fuel ethanol in the European Union. Over 1.2 billion liters of output were expected to be produced in the nation in 2022, an increase of almost 4% from the year before. In contrast, it was anticipated that Germany’s ethanol production would total 759 million liters in 2022. In the EU, Germany consumes more ethanol than any other country, with France coming in second [64]. The three types of feedstock used in the production of bioethanol include sucrose-containing feedstock, such as sweet sorghum, sugar beet, and sugarcane, starch-based feedstock such as maize, wheat, and barley, and lignocellulosic biomass, like straw, grasses, and wood [60,65].

## 6. Processes Involved in Bioethanol Production from Lignocellulosic Biomass

The recalcitrance of plant cell walls due to their complex nature poses some challenges with the use of lignocellulosic biomass for the obtainment of maximum ethanol yield. Therefore, in order to facilitate ethanol productivity and lower production costs, lignocellulosic materials are subjected to different stages of ethanol production processes, which include effective pretreatment processes, hydrolysis, fermentation, and distillation to separate ethanol produced from co-products [66] (Figure 3).

### 6.1. Pretreatment

Pretreatment is an essential stage in bioethanol production from lignocellulosic plant biomass as it aims at altering the complex structure of the material by breaking down the lignin seal to solubilize hemicellulose, reduce the crystallinity, and increase the porosity of the material so as to enhance the accessibility of hydrolyzing agents (enzymes or chemicals), which break down cellulose polymers into simple fermentable sugars [67]. An ideal pretreatment should be inexpensive, effectively de-lignify substrate materials, prevent the loss or deterioration of carbohydrates, produce high sugar yield, and prevent the formation of sugar degradation products [68]. Certain plant biomass, such as cassava, contains high cyanide content [69], which could affect enzymes and microorganisms and ultimately lead to low production of reducing sugars. Hence, Mohammed et al. demonstrated that 24 h of soaking and 120 min of the boiling pretreatment condition is able to reduce the cyanide content in cassava peel waste and improve the total recovery of carbohydrates [70].

In a bid to overcome the challenges inherent to the use of lignocellulosic materials, there has been a shift in pretreatment procedures starting from chemicals and heating methods to biological methods [71], but there is still no satisfactory result from the different pretreatment methods adopted so far in terms of technology for industrial large-scale production, cost-effectiveness, and production of a lower amount of inhibitory products [72]. Several pretreatment processes for lignocellulosic biomass have been proposed and practicalized. They include physical, chemical, physicochemical, and biological pretreatment processes, as shown in Figure 4. The objectives, advantages, and disadvantages of these different pretreatment processes are outlined in Table 3.

### 6.2. Hydrolysis

After the pretreatment process, the next stage is the hydrolysis procedure. Hydrolysis is a chemical process that involves the use (addition) of water to break down polymers (e.g., cellulose) into monomers (e.g., glucose). It is a chemical reaction that breaks the chemical bonds that exist between two substances and releases energy, in which one molecule of a substance receives an H^+^ ion while the other molecule obtains an OH^−^ group. Hydrolysis is needed to obtain simple fermentable sugars. Acids (HCl, H_2_SO_4_, etc.) and enzymes are commonly used catalysts in the hydrolysis of biomass.

#### 6.2.1. Acid Hydrolysis

Acid hydrolysis can be achieved by inorganic acids (liquid acid catalyst or solid acid catalyst) or organic acids. Liquid acid hydrolysis is of two types, viz., dilute and concentrated, each of which possesses unique characteristics on biomass. Two reactions are involved with the use of dilute acid hydrolysis. One involves the conversion of cellulose into sugar while the other involves the conversion of sugar into chemicals, and many of these chemicals act as growth inhibitors to fermenting organisms. Concentrated acid hydrolysis, which consists of about 70% acid content, operates at low temperatures (37.8 °C) and pressure. However, dilute acid is the most preferable in terms of economics and effect on biomass [79]. Solid (heterogeneous) acid catalysts have, in recent years, experienced an increase in their use for cellulose hydrolysis into glucose. Examples of these catalysts include H-form zeolites, functionalized silica, immobilized ionic liquids, metal oxides, supported metals, acid resins, heteropoly acids, carbonaceous acids, and magnetic acids [80]. Some of the advantages of heterogeneous catalysts are that they can easily be removed from a reaction mixture through a process like filtration. This is important for industrial manufacturing processes since it makes expensive catalysts simple, efficiently recoverable, and reusable [81]. They are also environmentally friendly and possess good thermal stability [82].

Heterogeneous catalysts also have some disadvantages. When the catalyst’s surface has been entirely covered by reactant molecules, the reaction cannot continue until the products have left the surface and some area has, once again, become available for a fresh batch of reactant molecules to adsorb or attach. This explains why the rate-limiting stage in a heterogeneously catalyzed process is frequently the adsorption step [81]. A heterogeneous catalyst is less active and selective compared to a homogeneous catalyst due to the possession of multiple active sites [82]. Also, solid catalysts have lower conversions than homogeneous catalysts and necessitate more extreme reaction conditions to provide the same conversions [83]. Organic acids, e.g., acetic acid, which are generally regarded as weak acids that do not dissociate completely in water, have been examined for the hydrolysis potential of biomass. For example, Kanlaya and Jirasak achieved a 30.36% yield of reducing sugars when 0.25 M acetic acid was used to hydrolyze cassava peels at 135 °C for 90 min [84].

#### 6.2.2. Enzymatic Hydrolysis

Cellulases are a class of enzymes used to catalyze the hydrolysis of cellulose. They are produced by bacteria or fungi. However, there is more interest in the use of cellulase produced by fungi than the ones produced by bacteria. This is because most cellulase-producing bacteria are anaerobes that have a very low growth rate. The release of monomeric sugars from cellulose requires the action of three groups of enzymes, viz., endoglucanases, exoglucanases (cellobiohydrolases), and β-glucosidases [60,85].

Endoglucanase: this is one of the enzymes of cellulose deconstruction that acts by splitting the polymer, i.e., the cellulose long chains into shorter molecules (which could be oligosaccharides or smaller polysaccharides units);Exoglucanase: this other group of enzymes frees/releases cellobiose (which is a disaccharide) from either the non-reducing end or the reducing end;β-glucosidase splits cellobiose and other short-chain cello-oligosaccharides into monomer units (glucose).

Other enzymes that have been used for the hydrolysis of plant biomass include xylanases that hydrolyze the major component of hemicellulose (xylan), amylases for the digestion of starch, etc. [86]. Xylan is a heterogenous/complex compound with a backbone consisting of β-1,4-linked xylosyl residues, and so, the xylanolytic enzymes generally consist of a collection of enzymes, such as endoxylanase, β-xylosidase, α-glucuronidase, α-arabinofuranosidase, and acetylxylan esterase [87], which act in collaboration to convert xylan into sugars [88]. Nevertheless, xylanases (endoxylanases) are the most crucial since they are directly involved in cleaving glycosidic bonds and releasing short xylo-oligosaccharides [87].

There are differences in characteristics that exist with the use of acid hydrolysis versus enzymatic hydrolysis, and these are outlined in Table 4.

State-of-the-art development in bioethanol production mainly focuses on technical advancement to obtain high ethanol yield, reduce the time and cost of processing, and, most importantly, minimize the number of steps involved in lignocellulosic biomass conversion. The technological approaches encompass a variety of techniques, such as process development, genetic modification of feedstocks, cellulase enzyme-based robust hydrolysis technique development, cell immobilization, recombinant microorganism development, fermentation under conditions of high solid load, solid-state fermentation development, and integration of different process steps, amongst others [90,91].

#### 6.2.3. Sugar Degradation Products/Fermentation Inhibitors

The treatment of hemicellulose by dilute acid results in the formation of toxic compounds, like furfural from pentose degradation and soluble aromatic aldehydes from lignin. On a weight basis, aromatic aldehydes are twice as toxic as HMF or furfural [92]. HMF is produced from the dehydration/degradation of hexose sugars during acid pretreatment or hydrolysis. These compounds inhibit the fermentation process, which is needed to produce valuable ethanol from sugar by entering the nucleus of the cell and getting attached to the replicating DNA, thus lowering microbial metabolism, reproduction, and enzymatic activities. Acid concentration, temperature, and time are important factors that determine the formation of inhibitors. High acid concentration and low temperature provide an optimum operating condition for acid hydrolysis of potato peels, as shown in Table 5 [93]. In addition to the above, the degradation of sugars also results in the formation of carboxylic acids, such as acetic, propionic, formic, and lactic acids [94]. The further degradation of furans gives rise to the formation of levulinic acid and formic acid and again, the contamination of substrate by microbes can lead to the formation of different acids, such as lactic acid [95]. The threshold of inhibition depends on the strain of microorganism and inhibitor tested [96,97] (Table 6 and Table 7). In the measurement of acid toxicity, pH is an essential parameter to be considered. The concentration of HMF beyond 8 g/L and the concentration of furfural beyond 5 g/L hinder the growth of all microbial strains. Notwithstanding, the concentration of furans at 1 to 2 g/L is lethal to the growth of some strains. Also, the growth rate can be inhibited at 15 g/L acetic acid concentration and 10 g/L formic acid concentration, but these concentrations have not been detected to cause a severe inhibitory effect on productivity [97]. In addition to the individual toxicity of compounds, it is also important to take cognizance of the cocktail effect of inhibitor products since the combined effect could elevate the toxicity of compounds. For instance, the interaction between acetic acid and furfural caused a negative effect, as a reduction in ethanol yield, specific growth rate, and biomass yield of *S. cerevisiae* were observed [98]. Furthermore, a more negative result was observed with the interaction of 2 g per liter of furfural and 2 g per liter of HMF than, with 4 g per liter of HMF and 4 g per liter of furfural acting separately [99]. Several measures can be taken to reduce the effect of fermentation inhibitors, viz., substrate concentration, including salts and produced ethanol, should be below the threshold tolerance of the microorganism involved, minimizing/preventing the use of procedures (e.g., chemicals) that lead to inhibitor formation at the time of pretreatment, in situ detoxification by microorganisms used in fermentation, and the modification of organisms either through microbial adaptation or genetic engineering [100].

### 6.3. Fermentation

Fermentation is a metabolic process that involves the breakdown of a substance into a simpler one by microorganisms such as bacteria, fungi, or yeast. In ethylic fermentation, it is a chemical process by which simple sugars are broken down anaerobically into ethanol.

#### 6.3.1. Industrial Fermentation Technology for Ethanol Production

The industrial fermentation technology applied in the production of bioethanol includes the following.

Batch fermentation is also referred to as a ‘closed system’ and is the most common and simplest method for producing ethanol. In this method, fermentation is carried out in separate batches. The fermenter is first loaded with the substrate, after which the microorganisms are added and left to ferment the substrate. Byproducts accumulate, which continuously changes the culture environment. The products are removed at the end of the fermentation process and the fermenter is cleaned and sterilized in preparation for the next round. The microbes in the fermenter show three distinct growth phases, viz., lag, log (exponential), and stationary phases. The batch fermentation method has some advantages, such as less labor demand, ease of operation, low investment cost, quick and easy control methods, complete sterilization, and less risk of contamination [102].Fed-batch fermentation is an improved version of the batch fermentation process. Here, the feeds containing substrate, culture medium, and other vital nutrients are loaded into the fermenter, after which the cultured microorganisms are introduced and left to ferment the substrate. The feed solution is continuously introduced into the fermenter on an incremental basis throughout the fermentation process without the removal of the products formed. The products are only removed/extracted at the end of each fermentation process. The amount of working volume is a limiting factor in this process [102].Semi-continuous fermentation is sometimes referred to as either repeated fed-batch fermentation or a combination of some features and is notable in the batch and continuous fermentation process. Here, the feed solution is loaded into the fermenter at a constant interval, and the products formed are removed intermittently (not regularly). This process usually requires fixed volume, i.e., the volume of fermented (used) medium removed from the fermenter is usually replaced by an equal volume of fresh feeds at a constant time interval. This practice could help to maintain the growth of microbes for some time, as they get to feed on freshly provided nutrients that replace the already exhausted ones and, also, the intermittent removal of formed products could prevent the fermenting organisms from quickly transiting into the inactive/death phase; hence, an increase in product yield could be achieved. This process allows for an extended fermentation time, and the cycle is not usually terminated until a decline in productivity is detected [103].Continuous fermentation, as the name implies, means that the feed solution is continuously loaded into the fermenting vessel and the products formed are constantly removed/extracted. This allows for a longer fermentation time; the cycle is not interrupted like it is in the batch fermentation process. The growth of microorganisms is, therefore, maintained for a long time in the fermenting vessel due to the fresh nutrient supply and the regular removal of products whose accumulation has been reported to be detrimental to fermenting microorganisms. Hence, this process results in higher productivity [104].

#### 6.3.2. Microorganisms for Sugar Fermentation

Ethanol production requires the fermentation of sugars present in biomass by various microorganisms.

Microorganisms for pentose sugar fermentation: pentose sugars, such as xylose and arabinose, have been regarded as ‘un-fermentable sugars’. From a broad perspective, this could be correct since most microorganisms, such as yeast, fungi, and bacteria, are unable to effectively utilize pentose sugars. Nevertheless, there exist certain strains that play a significant role in nature’s economy that have been reported to be capable of breaking down these five-carbon sugars [105].

Bacteria: the majority of filamentous fungi and yeast are unable to ferment pentose sugars anaerobically, but bacteria are able to convert xylose to ethanol under anaerobic fermentation [106]. Xylose-fermenting bacteria comprise both native and genetically modified strains. During xylose fermentation, bacteria do not form xylitol; instead, they use its enzyme, ‘xylose isomerase’, to convert xylose directly into xylulose, and xylulose is then converted into ethanol through the pentose phosphate pathway (PPP) and the Embden–Meyerhof–Parnas pathway [107]. Examples of pentose-fermenting mesophilic bacteria include *Aerobacter hydrophila*, *E. coli*, *Clostridium acetobutylicum*, *Bacillus polymyxa*, *B. macerans*, and *Klebsiella pneumonia* [108]. Thermophilic anaerobic bacteria have been suggested as promising candidates for the conversion of pentose sugars into ethanol. Some of the species that have been studied include *Thermoanaerobacter ethanolicus*, *T. brockii*, *T. thermohydrosulfuricus*, *Thermoanaerobacterium thermosaccharolyticum*, and *Thermoanaerobacterium saccharolyticum* B6A [109]. The benefit of utilizing bacteria, e.g., *E. coli* ATCC 11303 (pLOI297), for ethanol production is that the process does not need aeration to achieve high productivity, but the downside is the high possibility of contamination since it functions at higher pH. Other disadvantages include its high sensitivity to ethanol inhibition and loss of productivity due to plasmid instability in the course of prolonged operation. Successful large-scale application of bacteria in fermentation is not very certain compared to yeast [110].Yeast: yeast is a common and suitable organism for the production of ethanol from sugars. This microorganism has been reported to act favorably in the fermentation of hexose sugars compared to pentose sugars [111]. However, certain strains, such as *C. shehatae*, *Kluveromyces marxianus*, *P. tannophilus*, and *P. stipitis,* have been evaluated for their ethanol production potential. Several other species of yeast that are able to utilize the five-carbon sugar (xylose) include *Clavispora* sp., *Schizosaccharomyces* sp., and *Brettanomyces* sp. Also included are *Debaromyces species,* such as *D. nepalensis* and *D. polymorpha*, and *Candida species,* like *C. blankii*, *C. tenius*, *C. utilis*, *C. solani*, *C. tropicalis*, *C. parapsilosis*, and *C. friedrichii* [108]. Most yeasts are incapable of fermenting xylose directly, so they ferment/utilize xylulose, which is an isomer of xylose. The bacteria enzyme ‘xylose isomerase’ can catalyze the interconversion of xylose and xylulose (isomerization), which is achieved in a single step, whereas yeast utilizes xylose reductase to reduce xylose to xylitol and then makes use of xylitol dehydrogenase to convert xylitol to xylulose. Species of *Candida*, *Kluyveromyces*, *Brettanomyces*, *Torulaspora*, *Pachysolen*, *Saccharomyces*, *Hansenula*, and *Schizosaccharomyces* have been recognized as the best ethanol-producing yeast from xylulose [112]. Nutrient medium composition, temperature, aeration rate, and pH are some of the factors that affect xylose-fermenting yeast performance. Some of the benefits associated with the utilization of yeast, e.g., *P. stipitis,* for the conversion of xylose is that it has high selectivity for ethanol production, unlike bacteria and fungi, which form co-products with ethanol. It is also relatively tolerant to ethanol and low pH, properties that reduce the risk of bacterial contamination. However, the drawback of this organism (xylose-fermenting yeast) is that it requires a small amount of oxygen (≤2 mMol/L-h) to realize high conversion efficiency; it is relatively easy to achieve micro-aeration on the laboratory scale, but it is not easy to achieve in the industrial scale. Another downside of xylose-utilizing yeast is that it presents low volumetric productivities when compared to those obtained with bacteria or glucose-fermenting yeast [110]. Compared to *S. cerevisiae*, yeast that utilizes pentose sugars is poorly tolerant to ethanol, inhibitor products, and pH, and these attributes can result in low ethanol yield [113,114].Filamentous fungi: xylose conversion by fungi has not been extensively studied compared to xylose fermentation by bacteria and yeast [110]. Filamentous fungi, such as *Neurospora crassa*, *Mucor* sp., *Fusarium oxysporum*, *Monilia* sp., and *Paecilomyces* sp., have been known to have pentose sugar fermentation potential. One good thing about the fungal process is that it has the capacity to grow on natural plant material, which is usually absent in yeast-based processes. Nonetheless, the fungal system is associated with properties that make its application in ethanol production unpleasant, such as are low volumetric production, the longer time that it takes to ferment (4 days to 8 days), the small oxygen requirement, the high viscosity of fermentation broth, growth in large clumps instead of dispersed single cells, the co-production of acetic acid alongside ethanol as a major end-product, which ultimately leads to reduced ethanol formation, and low tolerance to substrate and product [108].

An examination of bacterial, yeast, and fungal xylose fermentation, in general, is presented in Table 8.

Microorganisms for hexose sugar fermentation: the species of microorganisms that are able to ferment hexose (e.g., glucose) are more than those that have pentose fermentation potential. Usually, microorganisms, e.g., *E. coli*, utilize glucose first until it gets exhausted during co-fermentation involving the mixture of sugars, before converting pentose sugars, e.g., xylose and arabinose, to ethanol. This sequential use of sugars can result in an incomplete or delayed consumption of secondary sugars, which, in turn, leads to a decrease in yield and productivity [115]. A wide range of microorganisms, such as bacteria, e.g., *E. coli*, *Zymomonas mobilis*, *Aerobacter hydrophila*, *Clostridium acetobutylicum*, *Klebsiella pneumonia*, *Bacillus polymyxa*, *B. macerans*, *Thermoanaerobacter ethanolicus*, *Clostridium thermosulfurogenes*, *C. thermocellum*, *C. thermosaccharolyticum*, *C. thermohydrosulfuricum*, fungi, e.g., *Mucor indicus*, *Neurospora crassa*, *Fusarium oxysporum*, *Monilia* sp., *Paecilomyces* sp., and yeast, e.g., *S. cerevisiae*, *Pichia stipitis*, *P. angophorae*, *Candida shehatae*, *Kluyveromyces fagilis*, *K. marxianus*, *Pachysolen tannophilus*, etc., have been used in the fermentation of hexose sugars [108]. But of all microbes that have been employed in the production of ethanol from plant biomass, *S. cerevisiae* is the most famous and commonly used in industrial-scale applications due to its high tolerance to ethanol, wide range of pH, high productivity, and ability to ferment a wide range of sugars [116]. Pyruvate is the first stage in the alcoholic fermentation pathway, and it is obtained by yeast (*S. cerevisiae*) through the Embden–Meyerhoff–Parnas (EMP) pathway, and bacteria (*Zymomonas*) are formed through the Entner–Doudoroff (ED) pathway. The next stage involves the decarboxylation of pyruvate to acetaldehyde; this reaction is catalyzed by the enzyme called pyruvate decarboxylase. The redox balance of alcoholic fermentation is realized via the reproduction of NAD^+^ when acetaldehyde is reduced to ethanol by the enzyme alcohol dehydrogenase. Alcoholic fermentation produces 1 mol of ATP through the ED pathway or 2 mol of ATP through the EMP pathway for each mol of glucose oxidized [117].

### 6.4. Distillation

Distillation is the process of purifying a liquid by separating the components of the liquid mixtures through heating/boiling and condensation. It is an effective purification technique that employs the differences in volatilities of constituents in a mixture [118]. Ethanol and water are soluble in each other, so distillation is required for the separation and concentration of ethanol from the fermentation broth. It is not possible to obtain 100% purity through simple distillation because of the azeotrope between water and ethanol; there is a strong hydrogen bond that exists between water and ethanol, which causes water to be attached to ethanol as it pulls out when heated and, therefore, about 95% of ethanol can be recovered through this process, which finds relevance in the solvent, chemical, cosmetic, and pharmaceutical industries [73]. To obtain 99.9% ethanol, i.e., anhydrous ethanol, further drying of ethanol or a dehydration step is required [37]. Irrespective of the product being recovered, distillation is an energy-intensive technique; it is expensive and consumes approximately 40% of the total energy used in the chemical and petroleum refining industries [119]. Notwithstanding, since distillation remains the main separation technique in process industries, it, therefore, becomes important to enhance its energy efficiency, particularly when applied in the separation of azeotropic mixtures. To this end, several special separation methods have been employed, such as liquid–liquid extraction, azeotropic distillation, pressure swing distillation, extractive distillation, pervaporation using membrane, salt addition, adsorption, etc. [120].

## 7. EU Legislation Supporting Advanced Biofuels

A 10% minimum goal for renewable energy used in the transportation sector was outlined in the 2009 EU Energy and Climate Change Package and was to be met by all EU member states in their respective nations by the year 2020. For the years 2010 to 2020, the Renewable Energy Directive (RED) outlined specific objectives and requirements for the transportation sector. The Renewable Energy Directive II (REDII) for 2021–2030 was approved by the European Union in 2018 [121]. It included a minimum target of 32% renewable energy consumption across all sectors and a reduction in greenhouse gases of at least 40%. Transportation should be decarbonized as a top priority going forward because, in comparison to an 18% fall or more in all other sectors, greenhouse gas emissions in the European transportation sector have decreased by only 3.8% since 2008. The EU has encouraged the development of advanced biofuels, which are made from non-food feedstock, as well as conventional biofuels, which are based on food, through the use of directives and national laws. The EU Renewable Energy Directive (RED), which was passed in 2009, stipulated that by 2020, 10% of the energy utilized in the transportation sector must originate from renewable sources [122].

The EU Indirect Land Use Change (ILUC) directive changed the RED in 2015 by placing a 7% cap on the amount that food/feed-based biofuels might contribute to the RES-transport target. A non-binding 0.5% goal for advanced biofuels in 2020 was included by the ILUC directive as an additional measure to encourage the use of biofuels made from non-food feedstock and wastes [123]. By agreeing to modify the Renewable Energy Directive (REDII) in June 2018, the EU Commission, Parliament, and Council established a target of 14% renewable energy sources—transportation energy and a sub-target of 3.5% for advanced biofuels by 2030. The capping of traditional food-based biofuels to a maximum of 7% of each member state’s 2020 level signifies a target of at least 7% for non-food-based/advanced fuels [122]. Advanced biofuels can be double counted towards the 3.5% target and the 14% target [123].

## 8. Industrial Projects/Technology on Advanced Bioethanol

There are several initiatives on the development of renewable fuels (bioethanol) from second-generation (2G) plant biomass, and most of these technologies follow certain process sequences, such as the pretreatment of raw biomass, the production of biocatalysts (enzymes/fermentation microbes, e.g., yeast), simultaneous saccharification and fermentation (SSF) of feedstock, simultaneous saccharification and co-fermentation (SSCF), or separate hydrolysis and fermentation (SHF) (Table 9). Two successful projects, POET-DSM Project LIBERTY in the USA [124] and GranBio in Brazil [125], exemplify the triumph over legal, technological, and financial obstacles in lignocellulosic bioethanol production. Through innovative processes and strategic partnerships, these ventures demonstrate the feasibility of large-scale bioethanol production from agricultural residues.

From the above, it can be deduced that the reviewed technologies employed steam pretreatment or chemical pretreatment, enzymatic hydrolysis, and wild or recombinant/genetically modified yeasts for the fermentation of sugars.

## 9. Conclusions

The different generations of biofuels have been reviewed, and it is obvious that the utilization of advanced biofuels obtained from second-generation and third-generation feedstock is more sustainable and cost-effective compared to biofuel generated from first-generation biomass. The processes involved in bioethanol production have been discussed; it was shown that the recalcitrance of lignocellulosic biomass compels it to undergo pretreatment steps to make the material more amenable to catalysis agents. It was also observed that although biological pretreatment and enzymatic hydrolysis of biomass are environmentally friendly, it takes longer process time and steps and could accrue high costs, for example, in the case of genetically modified organisms (GMOs). Also highlighted are the pentose and hexose sugar fermentation; here, it is worth noting that in a mixture of sugars, microorganisms consume hexose sugar (glucose) first. It is only after this sugar has been exhausted that it seeks pentose sugars. Furthermore, this study showed that the legislation in the EU supports the development of second-generation bioethanol production. It also observed the presentations of different projects on advanced bioethanol production. The pursuit remains on the search for the most optimal condition and technology for sustainable high ethanol yield with minimal cost. Therefore, for sustainable bioethanol production, the process optimization of abundantly available lignocellulosic materials, which are often considered waste biomass, e.g., cassava peels, sugar beet pulp, and *Ulva lactuca,* needs to be explored while employing an efficient yet environmentally friendly and cost-effective catalyst, like dilute acid, or a heterogeneous catalyst. This should be followed by hydrolysate fermentation using suitable microorganisms, such as *S. cerevisiae* and *P. stipitis* for hexose and pentose sugar, respectively. In addition to exploring different 2G bioethanol feedstock, comprehensive lifecycle analyses should be conducted to enhance the sustainability and economic viability of the biorefinery process.

## Figures and Tables

**Figure 2 molecules-29-01619-f002:**
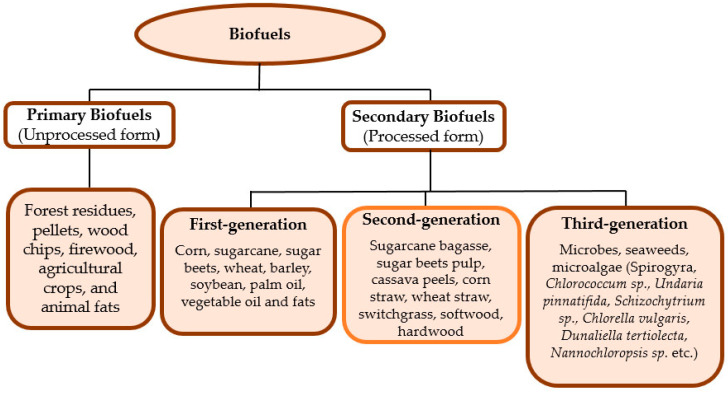
Biofuel classification and examples [34,35,36].

**Figure 3 molecules-29-01619-f003:**
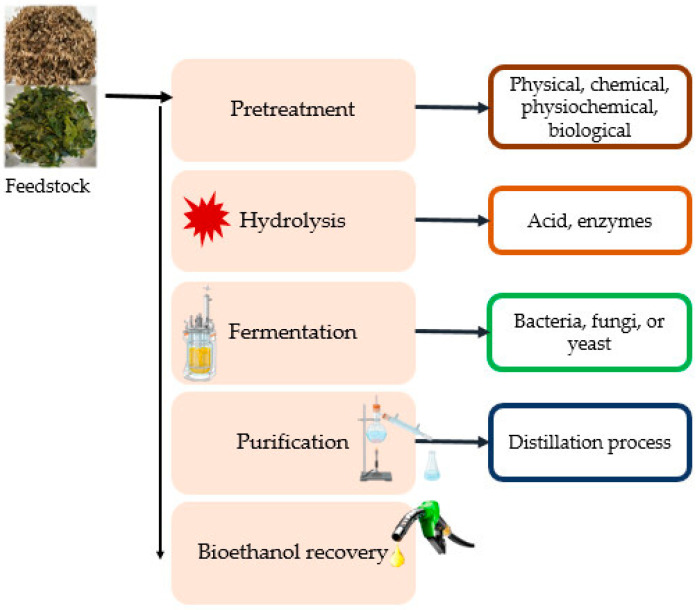
Stepwise process of bioethanol production from lignocellulosic biomass.

**Figure 4 molecules-29-01619-f004:**
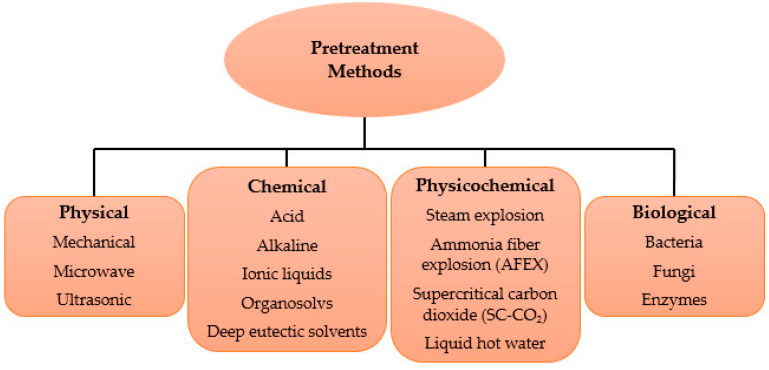
Different pretreatment processes.

**Table 1 molecules-29-01619-t001:** Compositions of selected lignocellulosic biomass.

Biomass	Cellulose (%)	Hemicellulose (%)	Lignin (%)	References
Oak wood	49.3	25.9	21.7	[22]
Sugar beet pulp	20.71	14.98	3.96	[23]
Sugarcane bagasse	50	25	25	[24]
Rice straw	34.6	27.7	17.6	[25]
Rice husk	33.4	22.1	22.8	[25]
Wheat straw	33.5	24.6	19	[25]
Oil palm empty fruit bunches	39.13	23.04	34.37	[5]
Corncobs	22.1	9.6	6.0	[26]
Banana rachis	26.1	11.2	10.8	[27]
Banana pseudostem	20.1	9.6	10.1	[27]
Cassava peels	9.05	7.50	9.16	[28]
Tygra hemp	50.82	27.79	14.68	[29]
Groundnut shell	35.7	18.7	30.2	[30]
Corn stover	36.1	21.4	17.2	[31]
Poplar	42.34	15.23	25.40	[32]
Waste from urban greening	22.96	6.86	22.73	[33]
Spring leaves	21.06	6.00	27.74	[33]
Autumn leaves	14.54	8.45	11.16	[33]
Jerusalem artichoke	25.99	4.50	5.70	[18]
Energy grass	37.85	27.33	9.65	[18]
Sunflower	34.06	5.18	7.72	[18]
Silage	39.27	25.96	9.02	[18]
*Miscanthus saccharifloris*	42.00	30.15	7.00	[18]
Reed	49.40	31.50	8.74	[18]

**Table 2 molecules-29-01619-t002:** Benefits and issues associated with secondary biofuels.

Secondary Biofuels	Benefits	Issues
First generation	Enhance energy, social, and economic security. Eco-friendly.	Impact on food security, land use, biodiversity, and carbon balances. High production cost. Partly blended with petroleum-based fuels.
Second generation	Lower impact on the food sector. Lower cost of feedstock. Enhance energy, social, and economic security. Better land-use efficiency. Eco-friendly.	Recalcitrance of cell walls. May incur a high cost of production. Infancy stage of cell wall polysaccharide biosynthetic machinery and its regulation.
Third generation	Mitigates greenhouse gases. Higher yields. Shorter harvesting cycle (1–10 days) High lipid content. Rapid growth rate. Reduced land use. Higher CO_2_ tolerance.	Algae require large amounts of water, phosphorus, and nitrogen to grow. Algae biomass requires dewatering before the extraction of lipids. Oil from algae tends to be more volatile (unsaturated), especially at high temperatures, and hence, more likely to degrade. Higher cost of cultivation. Higher energy consumption during harvesting.

**Table 3 molecules-29-01619-t003:** Advantages and disadvantages of the various pretreatment methods.

Pretreatment Methods	Objectives	Advantages	Disadvantages	References
Physical	To reduce biomass size and decrease crystallinity	Green pretreatment (rarely forms inhibitory product); improves hydrolysis rate	Energy intensive, not economically viable, and unable to remove/alter lignin	[73,74]
Chemical	To break down/solubilize/remove lignin and hemicellulose and increase surface area	Enzymatic hydrolysis might not be necessary (acid hydrolyzes lignocellulosic materials into simple sugars)	Corrosion of equipment, expensive, non-selective, requires high temperatures, chemical recovery issues, requires neutralization, and fermentation inhibitor problems	[60,75]
Physicochemical	To alter lignin, degrade hemicellulose, reduce cellulose crystallinity, and increase the surface area of biomass	Less use of chemicals, requires less energy compared to the mechanical method, high sugar recovery, limited environmental impact, and low cost	Unfinished disruption of lignin–carbohydrate matrix	[17,76]
Biological	To disrupt plant cell walls, selectively remove lignin, and degrade hemicellulose	Mild and eco-friendly, low energy requirement, and no formation of inhibitor byproducts	Relatively slow process and expensive (e.g., GMOs)	[77,78]

**Table 4 molecules-29-01619-t004:** Differences between acid hydrolysis and enzymatic hydrolysis [60,89].

Acid Hydrolysis	Enzymatic Hydrolysis
Corrosive	Non-corrosive
No specificity (selectivity)	More specific
Requires high process temperature (100 °C–160 °C)	Operates in low/milder conditions (44 °C–50 °C, pH 4.8)
Inhibitor formation issues	No inhibitor byproduct issues
Relatively low yield	Relatively high yield
In some instances after hydrolysis, requires neutralization with chemicals, which could be expensive (e.g., NaOH, KOH)	Initial high cost of enzymes. No neutralization needed
Not sensitive to operating conditions	Sensitive to operating conditions
Do not require genetic modification	Could necessitate the genetic modification of enzyme-producing organisms to improve hydrolysis
Non-environmentally friendly	More eco-friendly
Faster process (in minutes)	Takes longer process time (in hours)

**Table 5 molecules-29-01619-t005:** Effects of acid concentration, temperature, and time on inhibitor formation [93].

Acid Concentration (% *w*/*w*)	Temperature (°C)	Time (min)	Sugar Yield (g/100 g Biomass)	Inhibitor Concentration (g/100 g Biomass)	Ratio (Inhibitor: Sugar) (%)
5.0	135	30	26.32	0.6	2.25
5.0	150	15	25.97	2.2	8.4
10	135	8	55.2	1.1	1.9
10	150	8	46.4	1.91	4.1

**Table 6 molecules-29-01619-t006:** Growth (% of the control) of glucose and xylose-fermenting microorganisms in the presence of inhibitors (adapted from [101]).

Inhibitors	Concentration (g/L)	*S. cerevisiae*	*Z. mobilis*	*P. stipitis*	*C. shehatae*
Furaldehyde	0.5	53	82	75	81
	1	19	81	53	62
	2	10	44	1	9.7
Acetate	5	79	76	63	96
	10	52	44	64	84
	15	56	26	64	79
Hydroxymethylfuraldehyde	1	35	51	95	92
	3	17	69	31	32
	5	11	33	1.4	8
Vanillin	0.5	49	62	12	67
	1	14	37	0.7	9
	2	9	12	1.4	1.6
Hydroxybenzaldehyde	0.5	75	16	57	60
	0.75	47	8	30	23
	1.5	13	8	0	0.8
Syringaldehyde	0.2	100	82	72	89
	0.75	39	72	38	45
	1.5	19	60	3.6	5

**Table 7 molecules-29-01619-t007:** Concentration (g/L) of inhibitors at which the growth of microbes is completely hindered; σ_i_ (%) represents standard errors of the estimates (adapted from [96]).

Inhibitors	*S. cerevisiae*	σ_i_ (%)	*E. coli*	σ_i_ (%)	*B. subtilis*	σ_i_ (%)
Hydroxymethylfurfural	2.2	18.0	2.2	20.1	1.9	15.7
Syringaldehyde	2.5	8.2	2.7	13.7	2.0	6.0
Vanillin	1.08	22.9	2.2	12.0	1.84	18.3
2-Butanone	45.0	11.4	17.8	14.4	31.0	9.1
2-Butanol	36.0	12.6	21.0	6.5	20.0	18.7
Methyl propionate	23.0	11.6	13.68	13.4	21.0	6.0
Ethyl acetate	22.0	19.6	19.0	12.6	30.0	14.6

**Table 8 molecules-29-01619-t008:** Overall evaluation of xylose fermentation by bacteria, yeast, and fungi (adapted from [112]).

	1st Stage	2nd Stage	3rd Stage
Organism	Xylose to xylulose-5-P	Xylulose-5-P to pyruvate	Pyruvate to the final product(s)
Bacteria	Isomerization	Pentose phosphate + EMP pathway	Ethanol + mixed acids Ethanol + 2,3-butanediol Ethanol + acetone butanol
Yeasts	Oxidation reduction	Pentose phosphate + EMP pathway	Ethanol
Fungi	Oxidation reduction	Pentose phosphate + EMP pathway	Ethanol Acetic and lactic acids

**Table 9 molecules-29-01619-t009:** Projects/technology on advanced bioethanol production.

Projects/ Technology	Country/ Location	Feedstock	Technology Operation	Products/Production/Production Aim	References
Futurol^TM^ technology	France	Silvergrass (Miscanthus), agricultural residues, and wood residues	Steam explosion, on-site production of biocatalysts (enzymes and yeasts resistant to inhibitors, particularly acetic acid), enzymatic hydrolysis, co-fermentation (SSCF) of five-carbon and six-carbon sugars, and recovery of 2G ethanol, lignin, and stillage	55,000 tons (or 70 million liters of ethanol) of bioethanol	[126,127,128]
Sunliquid^®^ technology	Southwestern Romania Straubing, Germany (demonstration plant)	Wheat and other cereal straw	Chopping of feedstock into smaller sizes, steam explosion pretreatment, enzymatic hydrolysis, simultaneous fermentation with the yeast of both C_5_ and C_6_ sugars, ethanol, and vinasse recovery	50,000 tons of bioethanol on a yearly basis from 250,000 tons of agricultural residues Demonstration plant: 1000 tons of bioethanol from about 4500 tons of wheat straw, corn stover, etc.	[129,130]
Domsjö Fabriker	Sweden	Spruce and pine biomass (about 1.6 million cubic meters annually)	Debarking and chipping of timber logs, feeding into a digester alongside cooking chemicals. Combustion of the bark to generate energy in the form of steam. Washing, bleaching, and drying cellulose after cooking. Fermentation of dissolved hemicellulose and distillation to produce bioethanol, drying of refined lignin, and recycling of cooking chemicals to produce energy	Cellulose, lignin, and bioethanol, carbon dioxide processed into carbonic acid	[131,132]
ProEthanol2G project	Europe and Brazil	Wheat straw, sugarcane bagasse, and straw	Pretreatment and enzymatic hydrolysis to convert molecules into sugars, followed by fermentation with recombinant yeast strain of the sugar solution and distillation	Europe: bioethanol and electricity from 100% wheat straw Brazil: bioethanol, sugar, and electricity from 100% sugarcane crop, bagasse, and straw	[133,134,135]
BALI^TM^ Biorefinery Demo	Sarpsborg, Norway	Spruce, bagasse, willow, straw, wood, and energy crops	Chemical (sulfite) pretreatment, enzymatic hydrolysis, fermentation (conventional fermentation of C_6_ sugars, aerobic fermentation, or chemical conversion of C_5_ sugars), and chemical modification of lignin	Processing capacity of 1 to 2 MT per day of biomass Products: ethanol, lignin, and various chemicals	[136,137,138,139]
Bamboo biorefinery built on Chempolis’s patented formicobio^TM^ technology	Assam, Northeast India	Utilization of 300,000 tons of bamboo annually	Selective dissolution of biomass’s major components excluding cellulose by biosolvents under low temperature and pressure, purification of cellulose by washing with water, enzymatic hydrolysis of cellulose, fermentation, and distillation. Combustion of lignin-rich biofuel to produce steam and electricity	Production of 60 million liters of bioethanol, 19,000 tons of furfural, 11,000 tons of acetic acid, and 144 gigawatt hours of green energy, yearly	[140,141]
Crescentino biorefinery complex (PROESA^®^ proprietary technology)	Italy	Rice straw, wheat straw, and energy crops, e.g., *Arundo donax* (giant cane)	Characterization of energy crops, steam pretreatment, enzymatic hydrolysis and co-fermentation (SSCF), and valorization of secondary streams and co-products	Plant capacity—40,000 tons of bioethanol per annum from more than 200,000 tons of feedstock (dry mass) Generate about 13 MW of electricity from lignin	[142,143,144,145,146]
MARINER (Macroalgae Research Inspiring Novel Energy Resources) projects	United States (US)	-	Integrated cultivation and harvesting systems, advanced component technologies, computational modeling tools, aquatic monitoring tools, and advanced breeding and genetic tools	Production of seaweed (macroalgae) for biofuel production Estimated production of 500 million dry metric tons of macroalgae per annum, amounting to ~2.7 quadrillion BTUs (quads) of energy (liquid fuel) and ~10% of US yearly transportation energy demand	[147,148]
TATA project	India	Rice straw (design feedstock) and maize stalk (check case)	--	Bioethanol plant production capacity—100,000 liters per day	[149]
LignoFlag	Europe	Wheat straw, corn stover, etc.	Utilizes Sunliquid^®^ technology	Aims to increase plant production capacity to 60,000 tons of ethanol per annum and use co-products for energy generation and soil fertilization	[150]
IOCL’s (Indian Oil Corporation Limited) 2G Ethanol Bio-Refinery (Praj’s technology)	India	Rice straw	Acid and steam explosion pretreatment, enzymatic hydrolysis, co-fermentation with GMOs (genetically modified organisms) type yeast, distillation, and dehydration	30 million liters of ethanol from 200,000 tons of rice straw per annum	[151,152]
AustroCel’s bioethanol plant (Valmet’s automation technology)	Hallein, Austria	Waste materials from adjacent viscose pulp mill	Sulfite pulping/digestion of wood chips and fermentation of sulfite spent liquor (SSL) with yeast	30 million liters of bioethanol	[153,154,155,156]

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
