# Peer review of "Renewable Energy Potential: Second-Generation Biomass as Feedstock for Bioethanol Production"

_molecules, 2024, doi:10.3390/molecules29071619_

Round 1

Reviewer 1 Report

Comments and Suggestions for Authors

The review article delves into the crucial topic of bioethanol production from second-generation lignocellulosic biomass, underlining the significant challenges and promising opportunities in this field. It discusses the methods, techniques, and optimal conditions for generating biofuels from biomass and the intricate processes involved in bioethanol production. The importance of urgently addressing challenges such as biomass recalcitrance and associated costs to achieve large-scale production is underscored. However, it is imperative to note that there are still significant revisions to the manuscript to ensure its quality and clarity before publication.

General comments

1. No details on the methodology and search strategy are provided, nor are the inclusion/exclusion criteria or article selection process mentioned. This information is essential to assess the soundness of the methodology.

2. The presentation and discussion of the tables should be improved, and figures should be included to improve the visual understanding of the concepts presented in each item. Figures help illustrate the structure and composition of lignocellulosic biomass, bioethanol production processes, and biofuel classifications, among other aspects addressed in the review article (I recommend BIORENDER or something similar).

Specific comments

3. The abstract provides a good overview of the article. However, it needs vital details, such as the number and types of studies reviewed and the main findings.

4. The introduction could benefit from more contextualization of the current relevance of biofuels and the specific role of bioethanol derived from lignocellulosic biomass in the global energy transition.

5. Contemporary examples of the importance of bioenergy in different regions of the world could be added to the biomass and bioenergy chapter to illustrate its practical relevance.

6. In the chapter on the structure and composition of lignocellulosic biomass, the relationship between the structure and composition of lignocellulosic biomass and its recalcitrance could be more clearly detailed with visual or comparative examples.

7. In the chapter on biofuel groups, the environmental implications of each generation of biofuels could be further explored to highlight the specific advantages of second-generation biofuels.

8. In the chapter on bioethanol as an energy source, quantitative data on energy efficiency and reduced emissions associated with using bioethanol compared to fossil fuels could be included.

9. In the chapter on processes involved in bioethanol production from lignocellulose, emerging or innovative technologies in production processes could be explored to highlight recent advances in the field.

10. In the chapter on legislation, projects, and technology, specific examples of successful projects that have addressed legal, technological, and financial challenges in producing bioethanol from lignocellulose could be included.

11. In the conclusions, include specific areas for future research to address the challenges identified, such as new technologies, more efficient methods, or interdisciplinary approaches.

Comments on the Quality of English Language

Minor editing of English language required

Author Response

Dear Reviewers,

Thank you for the time dedicate to evaluate our work and to the valuable recommendations.

Please find our answers to your comments in the attached document.

Furthermore, we have highlighted the changes in the edited manuscript,

Best regards,

Sary AWAD

Reviewer 2 Report

Comments and Suggestions for Authors

Briefly: The title of the work does not correspond to its content.

An urgent problem today is the disposal of plant waste containing inedible components and the production of biofuels based on them. Therefore, the topic of the presented review may be relevant. However, the title and goals of the work do not correspond to its content. The authors mixed into one boiler the production of biofuel from plant biomass, which was rightly classified as secondary generation of biomass, and the use of microalgae for the synthesis of biodiesel, and the classical use of microorganisms for the biosynthesis of ethanol. It is necessary to either completely change the title, purpose of the work and introduction, or remove unnecessary sections. It is also necessary to understand the taxonomy of microorganisms so as not to confuse the different families and genera of fungi, bacteria and algae, micro- and macroalgae.

Author Response

(The authors gave the same response as above.)

Round 2

Reviewer 1 Report

Comments and Suggestions for Authors

Accept in current form.

Comments on the Quality of English Language

Minor English language editing required.

Reviewer 2 Report

Comments and Suggestions for Authors

Accept in present form